# Assessment of Psychiatric Symptomatology in Bilingual Psychotic Patients: A Systematic Review and Meta-Analysis

**DOI:** 10.3390/ijerph17114137

**Published:** 2020-06-10

**Authors:** Leire Erkoreka, Naiara Ozamiz-Etxebarria, Onintze Ruiz, Javier Ballesteros

**Affiliations:** 1Department of Neurosciences, University of the Basque Country UPV/EHU, 48940 Leioa, Spain; leire.erkorekagonzalez@osakidetza.eus (L.E.); javier.ballesteros@ehu.eus (J.B.); 2Department of Psychiatry, Galdakao-Usansolo Hospital, Osakidetza Basque Health Service, 48960 Galdakao, Spain; 3Department of Mental Health, BioCruces-Bizkaia Health Research Institute, 48903 Barakaldo, Spain; ruizonintze@gmail.com; 4CIBERSAM, 48940 Leioa, Spain

**Keywords:** psychosis, schizophrenia, bilingualism

## Abstract

Language plays an important role in psychiatric conditions. Language disturbances are core symptoms of psychiatric ailments, and language is the main diagnostic tool to assess psychopathological severity. Although the importance of language in psychiatry, the effect of bilingualism, and more specifically of using the mother language or a later acquired language at the time of assessing psychotic symptoms, has been scarcely studied and, thus, remains unclear. We conducted a systematic review and meta-analysis to ascertain whether differences exist in the severity of psychopathology in psychotic patients when assessed either in the mother language or in an acquired language. Of 3121 retrieved references from three databases (PsycINFO, MEDLINE, Embase) and complementary searches, four studies—including 283 psychotic patients—were included in the review. The meta-analytical combined effect suggested that more overall symptomatology is detected when clinical assessment is conducted in the mother language rather than in the acquired language (very low quality evidence, random effects model standardized mean difference (SMD) 0.44, 95% CI = 0.19 to 0.69, *p* value = 0.0006, I^2^ = 90%). Considering the growing migration flows and the increasing number of bilingual people in the world population, the effect of the chosen language at the time of conducting psychopathological assessments of psychotic patients is a clinically relevant issue. Based on our findings, we recommend that clinical interviews with bilingual psychotic patients should be conducted, when feasible, in the patient’s mother language.

## 1. Introduction

The absence of biological markers to diagnose psychiatric conditions [1] makes language the most important tool in the assessment of psychopathology. Apart from being the main means of assessment, language disturbances are core symptoms of psychiatric conditions [2]. Language and thought are intimately tied up, and a recurrent debate exists to determine whether it is mostly thought that influences language, or vice versa [3]. The principle of linguistic relativity, developed 60 years ago [4], proposed that the structure of a language affects the speaker’s thought and worldview, and in the past two decades, such principle has found a substantial degree of empirical support [5]. Whatever the relationship between both processes, there is no doubt that a connection exists between how we think and how we communicate those thoughts, and the language we use in communication will act as a bridge.

As we have mentioned, language is also the way mental health professionals explore patients’ inner thoughts, processes, and symptoms of psychiatric conditions, which are uniquely expressed through language, since we lack complementary tests, such as blood indicators or neuroimaging, with definite clinical meanings. Declarations and assessments of mental processes are verbal in nature [6], and thus, in the case of bilingual or polyglot patients, the chosen communication language could impact psychiatric diagnosis and treatment [7]. 

Because more than half of the population of the world is bilingual or polyglot [8], the effect of the language of choice at the time of conducting a psychiatric assessment is an important issue. Outside of places where several languages have coexisted for a long time, the increasing migration flows have also contributed to the growing number of world bilingual or polyglot population. Global estimates indicate that in 2015 there were about 244 million international migrants worldwide, or 3.3 per cent of the world’s population [9]. 

Language disorders are a key feature of psychosis. Studies have shown that schizophrenia is associated with deficits in language function, as well as structural and functional abnormalities in the regions of the brain involved in language perception and processing [10]. Premorbid language impairments have been associated with later schizophrenia diagnoses in population cohort studies [11], and while neuropsychological profiles of these patients demonstrate deficits across different cognitive domains, language processing and verbal memory appear to be specially affected [12]. The deficiencies in language include problems in speaking (flat intonation, unusual voice, unintelligible utterances); listening (inattention, distraction, failure of understanding); grammar (chaotic sentence structure and syntax, unusual order and sequence); vocabulary (limited vocabulary, neologisms, clanging and glossomania); as well as in reading (stilted prosody, word approximation, misunderstanding of idiom and metaphor) and writing (erratic handwriting, unusual use of size and space) [13].

There has been little research on the importance of language when assessing psychopathology and making a psychiatric diagnosis, and bilingualism is not a variable that is considered in everyday practice [14]. In most bilingual contexts, the tendency is to use the lingua franca, that is, the language most extensively used in a certain population, without considering the preference or the linguistic background of the patient. This assumes that bilingual patients will verbalize their symptoms equally in their mother or first language (L1) than in an acquired or second language (L2), though in the case of psychiatric conditions this is a hard statement to defend. Consequently, we aimed to conduct a systematic review and meta-analysis of the literature to ascertain whether differences exist in the severity of psychopathology expressed in L1 vs L2 in bilingual schizophrenic or psychotic patients.

## 2. Materials and Methods

### 2.1. Type of Studies

We planned to include randomized and non-randomized comparative studies with a between-groups design (parallel studies that allocated patients to diagnostic interview in L1 or L2) or within-group design (crossover studies that assessed each patient twice, one in L1 and another in L2). We prespecified to exclude case reports; cross-cultural and psychometric validation studies; studies focusing on linguistic or semantic issues, or cognition studies; and second language acquisition studies.

### 2.2. Search Methods for Identification of Studies

We searched PsycINFO, MEDLINE, and Embase with OVID from date of inception to March 2019. We did not use language, date, document type, or publication status limitations for searching studies. To retrieve as many potentially eligible studies as possible we used a sensitive but poorly specific search strategy: [(“mother tongue” OR “native language” OR “second language” OR “acquired language” OR bilingual) AND (psycho * OR schizo *)].

We complemented the electronic search with scanning references from included studies, and with a snowball search for studies quoting two primary key references in the literature of bilingualism and psychiatric symptomatology [15,16].

### 2.3. Data Collection and Analysis

To identify studies meeting eligibility criteria, titles and abstracts of retrieved studies were independently screened by two review authors (L.E., N.O.). All potentially eligible studies were then assessed full text for inclusion in the review by the same reviewers. Any disagreement over eligibility of particular studies was resolved through discussion with a third reviewer (J.B.).

We extracted psychopathology in psychotic patients assessed with standardized interviews or questionnaires; either a combined score for psychopathology or particular scores for psychiatric symptoms by psychopathological domains or factors, as reported on the original studies.

We provide a narrative synthesis of the findings from the included studies and, if data allowed, we combined individual studies estimate of effect in a weighted meta-analytical average. We used the mean difference (MD) as the effect size to extract and combine when outcomes were presented in the same metric; otherwise, we used the standardized mean difference (SMD). We interpreted SMD as a small effect, medium effect, or large effect according to Cohen’s criteria: 0.2 represents a small effect, 0.5 represents a moderate effect, and 0.8 represents a large effect [17]. We also used, as a complementary interpretation, the common language effect size (CLES) that represent the probability that a randomly selected observation from one group will be larger than a randomly selected observation from another group. As the no effect of SMD = 0 defines a CLES of 0.5, CLES—0.5 defines the probability of one group to be superior to the other [18]. Primarily, we used a fixed effect model to combine effect sizes to preserve the weighting of the studies with disparate sample sizes [19]. However, we also used a random effects model to ascertain the extent and nature of discrepancies, if any, between the average results produced by both models. We assessed heterogeneity with the I2 statistic to quantify inconsistency between studies. The I2 is interpreted as the percentage of variability that is due to heterogeneity rather than to sampling error or chance [20]. We interpreted heterogeneity according to I2 values and a visual assessment of overlapping confidence interval (CI) for individual effect estimates. We considered I2 values >50% presenting without overlapping CI as substantial heterogeneity and conducted sensitivity or influential analysis—based on leaving out a study at the time from the set of included studies—to inform of the impact of a particular study on between-studies heterogeneity and on the overall effect estimate. We present the overall quality of evidence according to the Grading of Recommendations Assessment, Development and Evaluation (GRADE) approach [21]. All analyses were performed with the R system using the packages “meta” and “metafor” [22,23,24]. 

A protocol for this review was registered in PROSPERO CRD42019122071 (available at: www.crd.york.ac.uk/PROSPERO/display_record.php?ID=CRD42019122071). The Appendix A presents the differences between registered protocol and this study.

## 3. Results

### 3.1. Search Results

We retrieved 3121 references after deduplication, of which 3105 were considered irrelevant. We assessed 16 full-text articles (corresponding to 13 independent studies) for eligibility. A total of 12 articles (corresponding to nine studies) were excluded. Seven studies presented outcomes that were non-prespecified: three articles—corresponding to one study—reported linguistic analysis [25,26,27] and four did not report overall or particular scores for psychiatric symptoms [28,29,30,31]. Three studies presented inappropriate study designs, case reports [15,32,33], and two studies presented wrong interventions focusing on cross-cultural studies [34,35]. Finally, four studies were included in the meta-analysis [16,36,37,38]. Figure 1 shows the flowchart of the review.

### 3.2. Characteristics of Included Studies

Table 1 presents the main characteristics of the included studies. The studies presented both paired and unpaired designs, and different psychopathological scales and metrics to combine (18-and 24-item BPRS; Bannister-Fransella Grid Test of Schizophrenic Thought Disorder; total BPRS score; average BPRS score), which justified the use of the SMD as the effect size since its standardization makes the SMD comparable across different measures and metrics.

Two studies each presented a within-group design [16,38], or a between-groups design [36,37]. Only one study randomized patients to the interview intervention [37]. Whereas Marcos et al. [16] showed that patients presented greater symptomatology when interviewed in L2, the studies of Milun et al. [38], Malgady and Costantino [37], and Brown and Weisman de Mamani [36] showed a greater score for psychotic symptoms in patients when interviewed in L1.

### 3.3. Meta-Analysis

Based on four studies including 283 patients, there is evidence of rating psychotic symptoms more severely when patients are interviewed in L1 than in L2. The fixed effect model estimate is small to moderate (SMD = 0.44; 95% CI = 0.19, 0.69; *p* value = 0.0006). However, heterogeneity between studies is extremely large (Q-test = 28.76, *p* value < 0.0001; I2 = 89.6%). As a consequence, the random effects model estimate, which accounts for between studies heterogeneity beyond chance, is small and non-significant (SMD = 0.12; 95% CI = −0.80, 1-03; *p* value = 0.80) (Figure 2, upper panel).

The single most influential study is Marcos et al. [16], which showed an extremely large effect with a direction of effect at odds with the pattern of the rest of the studies. Its deletion from the meta-analytical set reduces between studies heterogeneity (I2 = 62.0%), with a combined estimate supporting a moderate to important effect for greater symptoms when participants are interviewed in L1 both for the fixed effects model (SMD = 0.57; 95% CI = 0.32, 0.82; *p* value < 0.0001) and the random effects model (SMD = 0.69; 95% CI = 0.23, 1.16; *p* value = 0.0036). Heterogeneity is now moderate (Q-test = 5.26, *p* value = 0.072) and based on three studies that show the same direction of effect and overlapping confidence intervals, which makes inconsistency not a worrisome issue (Figure 2, lower panel).

Overall, the quality of evidence for the meta-analysis is very low (when including all studies) or low (when excluding Marcos et al. [16]) because of risk of bias (most studies do not present randomization to interview language or interview periods), inconsistency of estimates—heterogeneity—with the complete set of studies, and imprecision of estimates (low sample sizes and large CI supporting from small/moderate to large effects). If we include the study of Marcos et al. [16], the cumulated evidence is very uncertain about the effect of interviewing patients with psychotic symptoms either in their mother or acquired language. The probability of detecting more symptoms in patients interviewed in their mother language ranges from a small 12% (fixed effect model) to a negligible 3% (random effects model). However, if we exclude the study of Marcos et al. [16] the evidence suggests that interviewing patients with psychotic symptoms in their mother language slightly increases the detection of those symptoms. In this case, the probability of detecting more symptoms in patients interviewed in their mother language is 16% (from CLES for fixed effect model) to 19% (from CLES for random effects model); a difference we considered to be of clinical relevance. Additionally, if we focus all evidence from the only randomized study [37] the SMD suggests a large difference favoring the greater detection of symptoms in L1 over L2 (SMD = 1.18), with a 30% probability of detecting more symptoms.

## 4. Discussion

We found that the language chosen to conduct psychopathological assessments of bilingual psychotic patients might influence the severity of symptoms expressed. Psychotic patients seem to verbalize more severe symptoms when assessed in L1 than in L2. Of particular interest is the unexplained discrepancy of the results reported by Marcos et al. [16] with those of the rest of the studies included in our review. This discrepancy is difficult to explain. Looking at the individual values reported in the article, it seems that two participants scored more in the English Wechsler Vocabulary than in the Spanish Wechsler Vocabulary, which is at odds with the fact that Spanish was the participants’ native language. However, deleting those two extreme cases does not substantially modify the combined effect size (SMD changes slightly from −2.45 to −2.31). Additionally, there is an unexpected moderate negative correlation between the difference in BPRS scores and the difference in Wechsler Vocabulary Score (r = −0.41). Another influential study is Brown et al. [36], which presented a large weight in the combined result and included a preference language interview that could have biased the combined result upwards. However, when deleted, in the influence and sensitivity analysis the combined effect size does not decrease, as could be expected, but rather increases (SMD from 0.44 to 0.55). Both Marcos et al. [16] and Brown et al. [36] presented a large influence on the summary of evidence and lead to a cautious interpretation of overall results.

Besides the four studies fitting our eligibility criteria, other studies, mostly based on clinical observation or case reports, have also explored the issue of expressing severity of psychiatric symptoms in bilingual patients, and their results tend to support our findings. Del Castillo [15] presented a series of case reports of patients who reported more severe psychotic symptomatology in L1 than in L2. Gonzalez [28] described more hostility in L1 compared to L2 in a self-administered checklist. Price and Cuellar [6] also described more severe psychotic symptoms in L1, pointing to verbal fluency, acculturation, and self-disclosure as significant predictors of the differences. Southwood et al. [39] described more psychotic features, including prominent speech and thought disorder in L1 compared to L2. Finally, differences have also been detected at the time of diagnosing bilingual patients, with a specific risk of misdiagnosing psychotic disorders when the assessment is not conducted in the patient’s L1 [30].

The neural basis of psychosis is yet to be fully elucidated, and the brain processing of language in bilinguals is not completely understood [40]. Nevertheless, some authors propose that overlapping brain regions may be involved in L1 processing and psychotic symptoms, whereas L2 language production may involve greater prefrontal activity and a more widespread neural network beyond the language areas in the temporal lobe [36], depending on other factors such as fluency and age of second language acquisition, which have been described to influence language preservation in bilinguals with other conditions such as aphasia [41]. According to Brown and Weisman de Mamani [36], L1 production may activate psychogenic regions in the left temporal lobe, repeatedly associated to positive symptoms of schizophrenia, whereas symptoms could present as less severe in an acquired language if a more diffuse network is involved. This proposal may help to explain our findings. However, the independence between L1 and L2 neural circuitries is far from being established. Studies on aphasia have shown that two languages may share underlying neural circuitries for some linguistic processes, such as phonology, grammar, or semantics, but not for others [7]. The degree of neural overlap between both languages seems to depend on the degree of proficiency in L2 and on the age of acquisition [41,42,43]. In simultaneous bilinguals, that is, those who acquired both languages at the same time and early in life, and in high proficient bilinguals, brain activation is similar in the processing of both languages, mainly in the frontotemporal, temporoparietal, and occipital regions [43,44]. However, in low/moderate proficient bilinguals, mostly sequential bilinguals who acquired the second language later in life, L2 processing involves smaller and more widely distributed areas than L1, with a greater involvement of the right hemisphere [44]. This is consistent with the view that low proficient bilinguals recruit additional areas to compensate for the reduced proficiency, especially right frontal areas, which points out to a more effortful processing in L2 [45].

This study has several limitations: the small number of eligible studies to include in the meta-analysis, the low quality of evidence because of non-randomized designs and the presence of extreme heterogeneity of results due to the study by Marcos et al. [16]. Additionally, few languages have been studied in comparative studies—Spanish/English and Afrikaans/English—whereas non-comparative evidence from case studies presents more language diversity. Overall, these limitations make the quality of evidence from the review be low or very low, and it leads to a cautious interpretation of results. Most studies have not controlled for the fluency of the patients in each language. A greater fluency in L1 could explain why patients verbalize their symptoms in a wordier manner in L1 just as a matter of convenience, with no further neurobiological reasons explaining it. However, other authors [27] have described more dysfluency in L1 compared to L2 among psychotic patients, with a greater presence of linguistic markers of schizophrenia in L2, though these results have not been replicated.

To conclude, our study synthesizes the comparative evidence concerning severity assessment when conducting clinical interviews with psychotic patients in bilingual settings. It is based on few studies (four studies, 283 participants) with low or very low quality of evidence. The overall conclusion is that interviewing patients in L1 instead of L2 might improve the assessment of severity of psychiatric symptoms, or vice versa, that interviewing patients in L2 instead of L1 might result in an underestimation of the severity of psychiatric symptoms. Accordingly, the election of the language to interview bilingual psychiatric patients could have an impact on diagnosis and treatment plans. Apart from settings where multiple languages have historically coexisted, the growing migration flows will lead mental health professionals to face idiomatic challenges increasingly often, and as the language chosen to conduct psychopathological assessments seems to influence the expression and severity of symptoms, both the mother language and a patient’s preference should be considered when interviewing a bilingual patient.

On account of the low quality evidence, imprecision of estimates, small number of comparative studies retrieved thus far, and the importance of this topic for clinical practice, further randomized and powered designs comparing severity of symptoms in L1 and L2 with standardized measures in similar psychiatric populations are encouraged. Proficiency and fluency of patients in both languages should ideally be considered and included in further studies, in order to determine to what extent ease or convenience may account for the observed differences. Additionally, designs with reverse language direction (L2/L1) might inform and clarify possible linguistic problems underlying psychopathological expression.

## 5. Conclusions

In this study, a systematic review and meta-analysis was performed to determine whether differences exist in the severity of symptoms when psychotic patients are assessed using their mother tongue or an acquired language. The standardized mean difference (SMD) and the common language effect size (CLES) were used to data collection and analysis. The results suggest that psychotic symptoms might be more accurately assessed interviewing patients in L1 instead of L2, inasmuch as psychotic patients seem to verbalize more severe symptoms when assessed in L1 than in L2. The language chosen to interview bilingual psychiatric patients could also impact diagnosis and treatment plans. Moreover, migration phenomenon continues to grow and, in the future, mental health professionals will have to face idiomatic challenges increasingly often.

## Figures and Tables

**Figure 1 ijerph-17-04137-f001:**
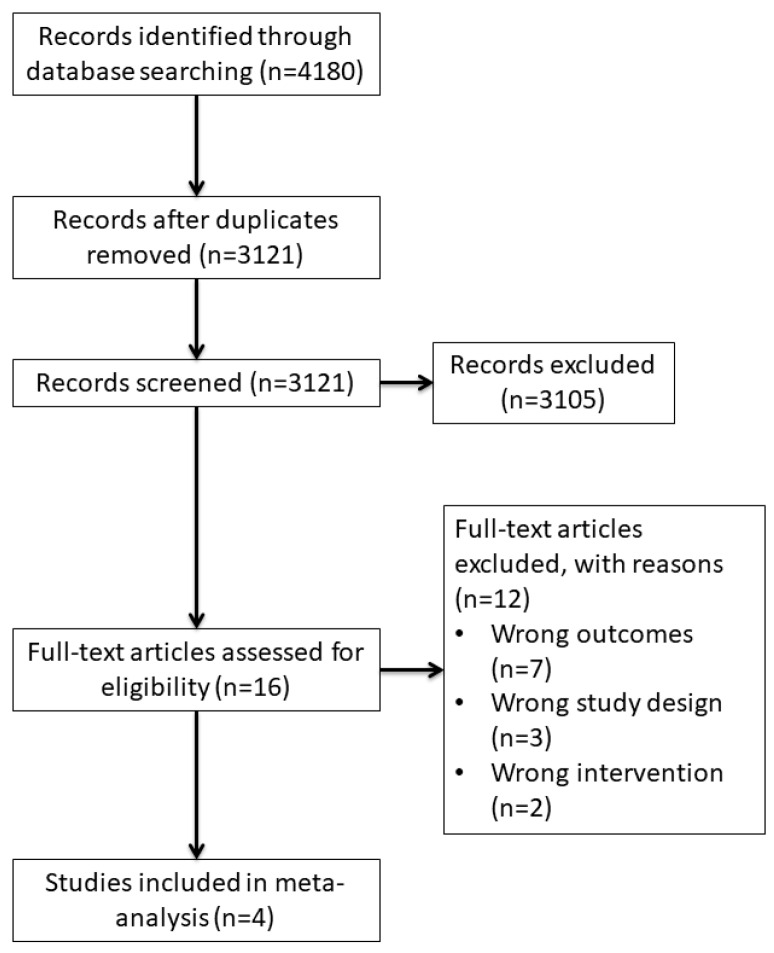
Flowchart of the review (*n* is the number of records/studies).

**Figure 2 ijerph-17-04137-f002:**
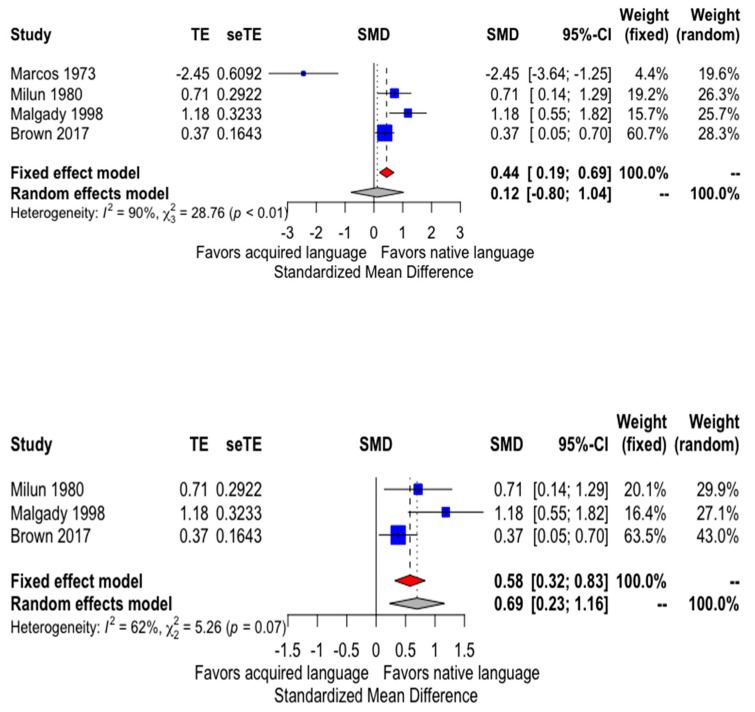
Symptomatic severity scores in bilingual psychotic patients interviewed in their mother or later acquired language. (**upper panel**) Results with the complete set of studies; (**lower panel**) results without the study by Marcos et al. [16]. SMD > 0 indicates more severity in the mother language.

**Table 1 ijerph-17-04137-t001:** Characteristics of included studies.

Study Country Official Language	Design. Participants Diagnosis	Gender Age Education	Outcome Scale	Languages (L1/L2)	Comments
Marcos et al., 1973 US English	Non-randomized within-group design *N* = 10 Schizophrenia	6 males/4 females Mean age 30.9 years (SD 6.3) Mean education 8.7 years (SD 2.2)	18-item BPRS	Spanish/English	Half of the patients participated first in one language interview and then in the other. Interviews were recorded with a closed-circuit television. Outcome data is available in Figure 1, and was extracted with WebPlotDigitizer (https://automeris.io/WebPlotDigitizer). SMD estimated from *t*-test paired analysis.
Milun et al., 1980 South Africa English	Non-randomized within-group design *N* = 10 Schizophrenia	Gender no reported Mean age 34.6 years (SD 14.5) Education no reported	Bannister-Fransella Grid test of Schizophrenic Thought Disorder	Afrikaans/English	Half of the patients participated first in one language interview and then in the other. SMD estimated from paired analysis, correlation between study sequences was not provided and estimated to be 0.70.
Malgady et al., 1998 US English	Randomized between-groups design *N* = 45 Schizophrenia	Data disaggregated by diagnosis no reported	18-item BPRS	Spanish/English	Participants with Spanish as L1 were randomized to Spanish diagnostic interviews (two conditions collapsed) or to English diagnostic interviews (two conditions collapsed). SMD estimated from unpaired analysis. The number of participants by condition is not provided, only total number of participants by psychiatric diagnosis. We have assumed equal allocation to conditions and restricted analysis to schizophrenia diagnosis (*N* = 45).
Brown et al., 2017 US English	Non-randomized between-groups design *N* = 218 Schizophrenia, schizoaffective disorder, other psychotic disorders	30.8 % female Mean age 41.05 years (SD 11.5) Most high school level or higher (94.5%)	24-item BPRS	Spanish/English	Assessment language was made by individual preference of interview (78% assessed in Spanish, 22% assessed in English). SMD estimated from unpaired analysis for the BPRS Thought Disturbance Scale. Number of participants by condition were estimated from reported percentages and total number of participants.

L1/L2: first language (L1)/acquired language (L2); *N*: number of participants; BPRS: Brief Psychiatric Rating Scale; SMD: standardized mean difference.

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
