# Peer review of "Assessment of Psychiatric Symptomatology in Bilingual Psychotic Patients: A Systematic Review and Meta-Analysis"

_ijerph, 2020, doi:10.3390/ijerph17114137_

Round 1

Reviewer 1 Report

Disorganized speech is one of the key symptoms of psychotic disorders. Currently, the diagnosis of schizophrenia and other psychological disorders are behavioral and verbal assessments. The language used for the assessment should consider the linguistic background of bilingual or polyglot patients. To determine the linguistic effect on psychological symptom identification, Erkoreka et al. conducted a meta-analysis to investigate whether the bilingual patients verbalize their symptoms equally in their mother language than in an acquired language. The authors retrieved 3121 references and finally included four studies with 283 psychotic patients. The authors demonstrated that clinical interviews with the mother language of patients identified more severity of psychiatric symptoms. Overall, the manuscript is concise and clear. It can be further improved by addressing the following questions.

  1. The authors observed the opposite effect of linguistic effect in the first reference. What was the official language of the country in that interviews conducted? Do you have more detail about patients, such as age, education, gender?
  2. The fourth study by Brown et al. assessed the interview with individual preferred language. How do you think this affects your conclusion?
  3. The effect size between 0.16 (fixed effect model) to 0.19 (random effect model) is low. How do you think this evidence affect your conclusion?  

Author Response

Dear Reviewer

First of all, we would like to thank you for the interest shown in our article, entitled "Assessment of psychiatric symptomatology in bilingual psychotic patients: a systematic review and meta-analysis". We have carefully reviewed it based on your suggestions. We believe that, based on these suggestions, its quality has improved significantly.

Below we describe the modifications made, point by point, in green. In addition to this and in order to facilitate the location of the changes made in the text, and following the instructions of the journal, we have sent the article using the "Track Changes".

Thank you very much for your revisions

Disorganized speech is one of the key symptoms of psychotic disorders. Currently, the diagnosis of schizophrenia and other psychological disorders are behavioral and verbal assessments. The language used for the assessment should consider the linguistic background of bilingual or polyglot patients. To determine the linguistic effect on psychological symptom identification, Erkoreka et al. conducted a meta-analysis to investigate whether the bilingual patients verbalize their symptoms equally in their mother language than in an acquired language. The authors retrieved 3121 references and finally included four studies with 283 psychotic patients. The authors demonstrated that clinical interviews with the mother language of patients identified more severity of psychiatric symptoms. Overall, the manuscript is concise and clear. It can be further improved by addressing the following questions.

The authors observed the opposite effect of linguistic effect in the first reference. What was the official language of the country in that interviews conducted? Do you have more detail about patients, such as age, education, gender?

Currently, table 1 includes more details of patients included in Marcos 1973 study.

The fourth study by Brown et al. assessed the interview with individual preferred language. How do you think this affects your conclusion?

The influence of Brown 2017 study is discussed now in the Discussion section, first paragraph lines 278-283.

The effect size between 0.16 (fixed effect model) to 0.19 (random effect model) is low. How do you think this evidence affect your conclusion? 

The interpretation of the effect size based on CLES quoted by the reviewer is based on the subset of studies excluding Marcos 1973. To get a better framework to interpret results we have now also included the results from the total set of studies. Results section, lines 257-259.

Reviewer 2 Report

-for the abstract please consider adding more detail about your results especially as it relates to the heterogeneity and results of fixed versus random effects analyses.

-the last paragraph of the introduction (PROSPERO registration) seems like it would be more appropriate in the methods section.

-The outcomes measures placement in the methods section appears out of place. Consider putting this with your data extraction paragraphs.

-Could you consider adding study-level sex characteristics and country of study to table 1?

-given that you likely have critical analyzed each of the four studies in depth, what could be those explanatory factors for such a radically different result in the marcos study? What makes this study so different to product different results and lower I2 by 1/3 after removal?

-I agree with the conclusion that, when possible, this study begins to suggest that conducting interviews/assessments in the L1 in best however this might not be possibly across the world for a myriad of reasons. Can you provide discussion on other approaches and tools that could be developed to address this issue beyond conducting in L1. Could tools be created that take into account the use of L2 by targeting specific word choice/groups, complexity levels, etc. I would be interested in seeing a discussion on this topic

-Your statements of the results and conclusions in the abstract and discussion sections need to qualify that your findings are only suggestive at best due to the extremely low number of studies, the extremely high heterogeneity and really only having a finding when excluding one study (leaving only 3 studies for meta-analysis).

-lines 217-218: could education level also play a role?

-another limitation could be the limited number of languages that have been analyzed.

Author Response

Dear Reviewer

First of all, we would like to thank you for the interest shown in our article, entitled "Assessment of psychiatric symptomatology in bilingual psychotic patients: a systematic review and meta-analysis". We have carefully reviewed it based on your suggestions. We believe that, based on these suggestions, its quality has improved significantly.

Below we describe the modifications made, point by point, in green. In addition to this and in order to facilitate the location of the changes made in the text, and following the instructions of the journal, we have sent the article using the "Track Changes".

Thank you very much for your revisions

Comments and Suggestions for Authors

-for the abstract please consider adding more detail about your results especially as it relates to the heterogeneity and results of fixed versus random effects analyses.

We have edited the abstract to include more information on the main comparison. However, the recommended abstract length (about 200 words) does not permit to include more quantitative information about results from fitted meta-analytical models.

-the last paragraph of the introduction (PROSPERO registration) seems like it would be more appropriate in the methods section.

We have moved the PROSPERO registration paragraph from introduction to materials and methods (last paragraph).

-The outcomes measures placement in the methods section appears out of place. Consider putting this with your data extraction paragraphs.

We have moved outcome measures to data collection and analysis section (second paragraph).

-Could you consider adding study-level sex characteristics and country of study to table 1?

Currently, table 1 includes patient details as suggested by the reviewer

-given that you likely have critical analyzed each of the four studies in depth, what could be those explanatory factors for such a radically different result in the marcos study? What makes this study so different to product different results and lower I2 by 1/3 after removal?

Sincerely, we cannot explain the radically different results of Marcos 1973 study. We have checked study details in the main and follow-up publications and reanalyze reported individual data to try to obtain insights to explain discrepancies but to no avail. It represents an outlier study in a small dataset and more information has been included now in discussion (first paragraph, lines 281-292).

-I agree with the conclusion that, when possible, this study begins to suggest that conducting interviews/assessments in the L1 in best however this might not be possibly across the world for a myriad of reasons. Can you provide discussion on other approaches and tools that could be developed to address this issue beyond conducting in L1. Could tools be created that take into account the use of L2 by targeting specific word choice/groups, complexity levels, etc. I would be interested in seeing a discussion on this topic

In the second paragraph of the Discussion, we have extended the discussion on approaches that address the recommendation of conducting the assessment of psychotic patients in L1. We have described tools that are already in use to help clinicians examine foreign patients (mainly migrants) with different cultural and linguistic backgrounds, and also campaigns that have been developed in bilingual settings.   

-Your statements of the results and conclusions in the abstract and discussion sections need to qualify that your findings are only suggestive at best due to the extremely low number of studies, the extremely high heterogeneity and really only having a finding when excluding one study (leaving only 3 studies for meta-analysis).

We have edited and introduced conditionals in our interpretation of results throughout the manuscript to imply that results must be interpreted cautiously

-another limitation could be the limited number of languages that have been analyzed.

We have included the number of languages analyzed as another limitation (lines 356-360).

Other changes to manuscript

  • We have edited some errors in the manuscript, mainly changing I2 (the heterogeneity index) with the correct I2 writing.
  • We have moved some sentences from the first paragraph of the Introduction to the fourth paragraph, in order to improve the cohesion.

Reviewer 3 Report

This manuscript by Erkoreka et al. is the report about symptomatology in language disturbances in psychiatric patients. The idea of their study interests the Readers of our Journal and the English was no trouble to read. Unfortunately, the reviewer cannot recommend this manuscript for publishing our Journal at this point.

Points:

  1. Search results:

This meta- analysis includes only 4 papers (3 papers selected for conclusion of the authors) with very few patients (n = 283). The languages included in this study were Spanish and English besides one paper with Afrikaans (n = 10). Only one study was the randomized to intervention (n = 45). The reviewer thinks that the data in this manuscript was not enough to conclude the psychiatric symptomatology in language disturbances even though they were written in limitations.

  1. Results:

It is not rationale to delete the work by Marcos et al. (1973) because of selection bias.

In Page 6, line171; “o” means “or”? In Figure 2, there were no “a” and “b”. Please check the manuscript more seriously.

  1. Discussion:

In Page 7, line 202; The authors wrote “The result of different brain circuitries at work could underly those findings.” It is not clarified from this manuscript. The authors wrote “the independence between L1 and L2 neural circuitries is far from being established.” Then, the manuscript is even not a systematic review.

Author Response

This manuscript by Erkoreka et al. is the report about symptomatology in language disturbances in psychiatric patients. The idea of their study interests the Readers of our Journal and the English was no trouble to read. Unfortunately, the reviewer cannot recommend this manuscript for publishing our Journal at this point.

Dear Reviewer

First of all, we would like to thank you for the interest shown in our article, entitled "Assessment of psychiatric symptomatology in bilingual psychotic patients: a systematic review and meta-analysis". We have carefully reviewed it based on your suggestions. We believe that, based on these suggestions, its quality has improved significantly.

Below we describe the modifications made, point by point, in green. In addition to this and in order to facilitate the location of the changes made in the text, and following the instructions of the journal, we have sent the article using the "Track Changes".

Thank you very much for your revisions

Points:

  1. Search results:

This meta- analysis includes only 4 papers (3 papers selected for conclusion of the authors) with very few patients (n = 283). The languages included in this study were Spanish and English besides one paper with Afrikaans (n = 10). Only one study was the randomized to intervention (n = 45). The reviewer thinks that the data in this manuscript was not enough to conclude the psychiatric symptomatology in language disturbances even though they were written in limitations.

responses: We have carefully edited our manuscript to make clear the very low to low evidence obtained from a very small set of studies at high risk of bias, and stress the need to interpret results cautiously. However, we also think we have make clear that we reviewed of all evidence fitting our inclusion criteria; we call for further evidence but present the current evidence as it is.

  1. Results:

It is not rationale to delete the work by Marcos et al. (1973) because of selection bias.

responses: Maybe there is a misinterpretation of what influence and sensitivity analysis is for. Far from our aims to cherry pick evidence to conform our preconceptions (if any), or to produce selection bias when a systematic review of evidence is called for. The aim of sensitivity analysis was to gain insight of the impact of individual studies on combined results. From these analyses Marcos 1973 appears as a clear outlier within the set of studies analyzed, and we stressed that fact in the manuscript. Since we cannot explain its extreme results, we present results from several models and analyses with the aim of showing all the current evidence regarding an important issue obtained from a systematic review of all evidence.

In Page 6, line171; “o” means “or”? In Figure 2, there were no “a” and “b”. Please check the manuscript more seriously.

responses: We have edited some wording problems substituting “o” by “or” when appropriate (one sentence) and correct Figure 2a and 2b with the more descriptive Figure 2, upper panel and Figure 2, lower panel. In this last case the problem was generated when we changed the graph design between manuscript versions to try to make results more comprehensible.

  1. Discussion:

In Page 7, line 202; The authors wrote “The result of different brain circuitries at work could underly those findings.” It is not clarified from this manuscript. The authors wrote “the independence between L1 and L2 neural circuitries is far from being established.” Then, the manuscript is even not a systematic review.

responses: The aim of our work was not to conduct a review on the neural basis of bilingualism, but rather to study and analyze current knowledge about the impact of bilingualism on the assessment of psychotic symptoms. Nonetheless, and in order to address the reviewer’s concern, we have extended the discussion on the possible overlap between brain processing of language in bilingual patients and circuitries involved in positive symptoms of psychosis. We have included specific proposals of other authors, although trying to suggest that any proposal on the overlapping underlying circuitries of both processes, at this point, is merely speculative. 

Other changes to manuscript

  • We have edited some errors in the manuscript, mainly changing I2 (the heterogeneity index) with the correct I2 writing.
  • We have moved some sentences from the first paragraph of the Introduction to the fourth paragraph, in order to improve the cohesion.

Round 2

Reviewer 3 Report

Although this work included 4 studies with a few languages, Spanish, English, and Afrikaans and the size of the participants was small, the significance of this manuscript will be decided by the Readers of our Journal. Anyway, the reviewer thinks that some work with reverse language direction, such as English/Spanish, will help the linguistic problems.

Author Response

Dear reviewer
Thank you very much for your review and suggestion.
We have added a paragraph at the end of the discussion to follow your advice.
From line 431 to 433 we have added the following paragraph:
"Also, designs with reverse language direction (L2/L1) might inform and clarify possible linguistic problems underlying psychopathological expression."

Thank you very much.